# Oral Tetracycline-Class Drugs in Dermatology: Impact of Food Intake on Absorption and Efficacy

**DOI:** 10.3390/antibiotics12071152

**Published:** 2023-07-05

**Authors:** Rachel E. Tao, Stuti Prajapati, Jessica N. Pixley, Ayman Grada, Steven R. Feldman

**Affiliations:** 1Center for Dermatology Research, Department of Dermatology, Wake Forest University School of Medicine, Winston-Salem, NC 27104, USA; 2Department of Dermatology, Case Western Reserve University School of Medicine, Cleveland, OH 44106, USA; 3Department of Pathology, Wake Forest University School of Medicine, Winston-Salem, NC 27101, USA; 4Department of Social Sciences & Health Policy, Wake Forest University School of Medicine, Winston-Salem, NC 27101, USA; 5Department of Dermatology, University of Southern Denmark, 5000 Odense, Denmark

**Keywords:** tetracycline-class drugs, tetracycline, doxycycline, minocycline, sarecycline, drug absorption, dermatology, acne, rosacea, hidradenitis suppurativa

## Abstract

Tetracycline-class drugs are frequently used in dermatology for their anti-inflammatory properties to treat skin diseases such as acne, rosacea, and hidradenitis suppurativa (HS). The American Academy of Dermatology (AAD) clinical guidelines do not offer guidance regarding the co-administration of food with tetracycline-class drugs. The objectives of this study were to review the available evidence regarding whether taking tetracycline-class drugs with food decreases systemic absorption and is associated with an impact on clinical efficacy. A literature search was conducted using the PubMed database between February to May 2023 using the keywords “tetracycline-class drugs”, “pharmacokinetics”, “absorption”, and “dermatology”. Inclusion criteria included articles written in English and relevant to the absorption and efficacy of tetracycline-class drugs. This search yielded 131 articles written between 1977 to 2022, of which 29 met the criteria for review. United States Food and Drug Administration (FDA)-approved prescribing information for oral formulations of tetracycline, doxycycline, minocycline, and sarecycline were reviewed. Systemic absorption of tetracycline decreased when co-administered with food. Systemic absorption of oral doxycycline and minocycline was variable with food co-administration. The impact on bioavailability varied with the drug formulation and dosage. The absorption of oral sarecycline decreased when administered with food. Sarecycline is the only oral antibiotic where population pharmacokinetic studies demonstrated limited or no impact of food intake on clinical efficacy. There are no available data for other tetracycline-class drugs in dermatology. If patients find it more tolerable to take doxycycline, minocycline, and sarecycline with food to avoid gastrointestinal distress, this may merit consideration to encourage patient adherence. Since the impact of food intake on absorption varied with the dosage form of doxycycline and minocycline, consulting the appropriate package insert may give clinicians additional insight into differences in the various formulations.

## 1. Introduction

A variety of pharmacokinetic measures are used to describe the absorption and efficacy of drugs. The bioavailability of a drug is the fraction of the substance that is absorbed and becomes available in the systemic circulation [1]. A drug’s bioavailability is influenced by the rate of administration, gastrointestinal absorption, and hepatic metabolism as well as the concentration administered [1]. Interactions between oral medications and food can enhance or diminish the bioavailability and therapeutic efficacy of the drug [2,3]. These changes can be studied through single-dose pharmacokinetic studies that examine alterations in the extent and rate of drug absorption [2]. The extent of absorption can be measured through the area under the curve (AUC) [2]. Absorption rate is measured through the peak concentration (C_max_) and the time to peak concentration (T_max_) [2]. The AUC represents the total drug exposure to the tissue across time and provides comparison between different formulations [1,2]. The AUC is the integral of the area under the curve of the drug plasma concentration graphed on the *y*-axis versus time following drug administration graphed on the *x*-axis [1]. The AUC is proportional to drug absorption [1]. The C_max_ is the highest concentration of a drug in the bloodstream or target organ after administration [4]. The T_max_ is the amount of time for a drug to reach the C_max_ after oral administration [5]. Even if food affects oral drug absorption, the effect is not always clinically significant [2].

Tetracycline-class drugs are frequently employed in dermatology for their anti-bacterial and anti-inflammatory activities and include tetracycline (first generation), doxycycline and minocycline (second generation), and sarecycline (third generation) [6,7]. The anti-inflammatory properties of drugs in the tetracycline class occur through the inhibition of bacterial products stimulating inflammation, suppression of neutrophil migration and chemotaxis, inhibition of T-lymphocyte activation and proliferation, inhibition of phospholipase A2 and matrix metalloproteinases, inhibition of mast cell activation, scavenging of reactive oxygen species, reduction of pro-inflammatory cytokine production, inhibition of granuloma formation in vitro, and inhibition of nitric oxide synthase expression [6,8]. These properties of tetracycline-class drugs make them useful for treating not only cutaneous infections but also acne, rosacea, and hidradenitis suppurativa (HS) [6,9,10].

Tetracycline-class drugs chelate with polyvalent metallic cations including iron, aluminum, magnesium, and calcium to form tetracycline-metal complexes that are poorly absorbed from the gastrointestinal tract [11]. Taking tetracycline-class drugs with milk, antacids containing polyvalent cations, and iron salts could reduce absorption by 50–90% [11]. The package inserts for tetracycline-class drugs state that medication absorption is impaired by antacids containing aluminum, calcium, or magnesium; bismuth subsalicylate; and preparations containing iron [12,13,14,15]. Although dermatologists typically counsel patients to take doxycycline with food to minimize side effects, it is not well understood whether taking tetracycline-class drugs with food decreases medication effectiveness. The American Academy of Dermatology (AAD) clinical guidelines for the management of acne (2016), rosacea (2019), and HS (2019) did not provide recommendations on the co-administration of food with systemic antibiotics [16,17,18]. Therefore, the objective of this study is to review the current evidence on the effects of food intake on systemic absorption and potential impact on clinical efficacy of tetracycline-class drugs used in dermatology. 

Tetracycline-class drugs are available in different oral formulations including immediate-release or extended-release and tablet or capsule formulations. Immediate-release formulations are absorbed and reach their peak level quickly [19]. Extended-release formulations are designed to last longer in the body, to reduce frequency of dosing, and to maintain steady levels of the medication [19]. Extended-release formulations are further classified into sustained- and controlled-release formulations [19]. Sustained-release formulations prolong drug release from a tablet or capsule while controlled-release formulations release the active ingredient at the appropriate rate to maintain drug levels in the body over a specific time period [19]. Drugs in capsule form are absorbed more quickly and tend to have greater bioavailability compared to tablets [20].

## 2. Results

### 2.1. Tetracycline

Tetracycline comprises 4, 6-carbon rings with a hydroxyl-group and a methyl-group attached at C-6 [21]. Oral tetracycline-class drugs are associated with gastrointestinal adverse effects including anorexia, nausea, vomiting, diarrhea, glossitis, dysphagia, enterocolitis, and inflammatory lesions in the anogenital region [22,23]. Tetracycline, doxycycline, and minocycline are broad-spectrum antibiotics that inhibit bacteria in the skin and gastrointestinal tract and can cause gut dysbiosis, increased susceptibility to infection, dysregulated metabolism, and infection by *Clostridium difficile* causing diarrhea and colitis [21,22,23].

Patient resources recommend that patients not take tetracycline with food [12]. The tetracycline package insert states that food and some dairy products interfere with absorption of tetracycline (Table 1) [12]. Patients taking tetracycline with high-carbohydrate and high-fat meals have reduced serum tetracycline concentrations to approximately half the tetracycline concentration from fasting patients [24]. Based on single-dose data, mean tetracycline serum concentrations during a regimen of 250 mg every 6 h would be approximately 4.5 µg/mL in a fasting patient [24]. In a patient taking the same tetracycline regimen with meals, mean serum concentrations would decrease to approximately 2.4 µg/mL, an insufficient level for activity against less susceptible organisms [24]. In another study, absorption of tetracycline was reduced by 81% when administered with iron, by 65% when administered with milk, and by 46% when administered with food [25].

### 2.2. Doxycycline

Doxycycline comprises 4, 6-carbon rings with a hydroxyl-group at C-5 and a methyl group at C-6 [21]. Doxycycline hyclate is a water-soluble salt form and doxycycline monohydrate is a less water-soluble salt form of the same drug [33]. Doxycycline can be taken with food or milk if gastric irritation occurs [22,23]. Doxycycline has been associated with an increased risk of esophagitis and inflammatory bowel disease [34]. Esophagitis occurs more frequently with doxycycline hyclate than monohydrate due to the acidity of the hyclate formulation [34]. 

When doxycycline hyclate delayed-release 100 mg tablets were administered with a high-fat meal including milk, the mean C_max_ was 24% lower and the AUC was 13% lower compared to fasting conditions (Table 1) [22,23]. When doxycycline hyclate delayed-release 150 mg tablets were administered with a high-fat meal including milk, the mean C_max_ was 19% lower and the AUC was unchanged compared with fasting conditions [22,23]. It is unclear whether the decrease in absorption has an impact on clinical efficacy [22,23].

In a previous study, the serum concentrations from patients taking doxycycline with meals were reduced by 20% compared to the serum concentrations in fasting patients [24]. In a fasting patient taking doxycycline, the mean serum level obtained during a multiple-dosage regimen of 200 mg/day would be approximately 4.4 µg/mL [24]. In a patient taking doxycycline with meals, the mean serum level would decrease to approximately 4.0 µg/mL, a sufficient level of antibiotic for activity against the most susceptible organisms [24].

When doxycycline hyclate tablets were administered with a high-fat meal including milk, the mean C_max_ decreased by 24% and the AUC decreased by 15% [26]. The clinical significance of these decreases in absorption is unknown [26]. When doxycycline hyclate capsules were administered with a high-fat meal including milk, the mean C_max_ decreased by 20% and the AUC was unchanged [26].

When doxycycline monohydrate was administered with a high-fat meal, the T_max_ was delayed by an average of 1 h and 20 min, but the C_max_ was increased by 7.5% and the AUC was increased by 5.7% compared to fasting conditions [27]. The doxycycline monohydrate formulation has a more basic pH and is more water soluble compared to doxycycline hyclate [34]. 

Administration of a doxycycline 40 mg modified-release capsule with a 1000 calorie, high-fat, high-protein meal including dairy products decreased the C_max_ by about 45% and the AUC by 22% compared to fasting conditions [28]. The package insert recommends that patients take this branded doxycycline at least 1 h prior to or 2 h after meals [28].

### 2.3. Minocycline

Minocycline comprises 4, 6-carbon rings with a dimethylamine group attached at C-7 that makes it one of the most lipophilic tetracycline-class drugs [21]. Minocycline can be taken with or without food [14,29,30,31,32]. When minocycline hydrochloride tablets were administered with a meal that included dairy products, the extent of absorption decreased by 6%, the C_max_ decreased by 12%, and the C_max_ was delayed by 1.09 h compared to fasting conditions (Table 1) [29]. When minocycline hydrochloride capsules were administered with a high-fat meal, the extent of absorption was unchanged and the mean T_max_ was delayed by 1 h compared to fasting conditions [14]. When minocycline hydrochloride extended-release tablets were administered with a high-fat, high-calorie meal that included dairy products, the C_max_ was 707 ng/mL compared with 700 ng/mL under fasting conditions, the T_max_ was 3.5 h compared to 2.0 h under fasting conditions, and the AUC was 12,000 ng.h/mL under fed conditions compared to 10,874 ng.h/mL under fasting conditions [30]. When serum concentrations of minocycline hydrochloride were compared when administered with milk, a meal, and 300 mg ferrous sulphate in two groups of eight volunteers, the mean AUC decreased by 27% with milk, 13% with food, and 77% with iron compared to fasting conditions [25]. Pharmacokinetic and pharmacodynamic studies on minocycline did not study the effect of food intake on clinical efficacy.

### 2.4. Sarecycline

Sarecycline comprises 4, 6-carbon rings with a 7-methoxy-methyl-amino-methyl-methyl at C-7 [8]. Sarecycline can be taken with or without food [15]. Sarecycline has narrow-spectrum activity compared to other tetracycline-class drugs and has little to no activity against Gram-negative bacteria, which reduces its effect on the microbiome [34]. In three sarecycline clinical trials, the only adverse gastrointestinal side effect occurring in at least 1% of subjects was nausea, which occurred in 3.1% of subjects [15]. 

Based on a bioavailability clinical study, when sarecycline is co-administered with a high-fat, high-calorie meal including milk, the C_max_ decreased by 31% and the AUC decreased by 27% (Table 1) [15]. In another study, data from 12 clinical studies of sarecycline were analyzed using population pharmacokinetic (PPK) modeling, exposure–response modeling, and pharmacodynamics. Co-administration of sarecycline with a high-fat meal reduced the exposure at steady state (AUCss) by 21.7% and was associated with a decrease in efficacy of just 0.9 inflammatory lesions (placebo-adjusted change). There were no clinically significant impacts on efficacy that necessitate dose adjustments [7].

### 2.5. Low-Dose Doxycycline and Minocycline Efficacy Data

Taking standard-dose tetracycline, doxycycline, and minocycline with food can lower absorption with an unknown effect on clinical efficacy. Low-dose doxycycline and minocycline may be effective for treating acne, rosacea, and HS (Table 2). 

When administered to adults with moderate facial acne, doxycycline hyclate 20 mg tablets twice daily for 6 months reduced the number of total comedones by 53.6% versus 10.6% in placebo group (*p* < 0.01), total inflammatory lesions by 50.1% compared to 30.2% in placebo group (*p* = 0.04), and total lesions by 52.3% compared to 17.5% in placebo group (*p* < 0.01) [35]. Another study compared adults with moderate facial acne who received doxycycline hyclate 20 mg tablets twice daily with patients who received doxycycline 100 mg tablets daily for 3 months [36]. Both groups of patients had decreased mean inflammatory lesion counts of papules and pustules, and the 20 mg twice daily dosing group had papules reduced by 84% and pustules reduced by 90% [36]. Although a statistical comparison between the two groups was not performed, patients in the 20 mg twice daily dosing group had a lower incidence of gastrointestinal adverse events [36]. 

When administered to patients with rosacea, doxycycline 40 mg daily reduced inflammatory lesion count at a serum concentration below that needed for antimicrobial effects [41]. In two phase III, parallel-group, multicenter, randomized, double-blind studies, treatment of adults with moderate-to-severe rosacea with doxycycline 40 mg controlled-release capsules once daily for 16 weeks reduced the total inflammatory lesions by 11.8 in one study and 9.5 in the other study compared with 5.9 and 4.3, respectively, in the placebo arms (*p* < 0.001) [37]. In another randomized, double-blind study, treatment of adults with rosacea with doxycycline monohydrate 40 mg capsules once daily for 16 weeks and metronidazole 1% gel for 12 weeks had a greater reduction in inflammatory lesion count from baseline (−13.44 vs. −6.5; *p* = 0.0006), investigators’ global assessment (IGA) score from baseline (−63.4% vs. −41%; *p* = 0.005), and erythema score from baseline (−1.3 vs. −0.7; *p* = 0.01) compared with topical metronidazole 1% gel monotherapy for 12 weeks plus placebo [38]. When combined with metronidazole 1% gel, doxycycline 40 mg delayed-release once daily was equally effective as doxycycline 100 mg once daily at improving rosacea [39]. When patients with moderate-to-severe rosacea received either doxycycline 40 mg delayed-release once daily or doxycycline 100 mg once daily for 16 weeks in addition to topical metronidazole 1% gel once daily, the mean change from baseline to week 16 in inflammatory lesion count was similar in both study groups and at all study visits [39]. 

In a randomized controlled trial in which patients with HS received doxycycline 40 mg modified-release once daily or doxycycline 100 mg regular-release twice daily for 12 weeks, Hidradenitis Suppurativa Clinical Response (HiSCR, a ≥ 50% reduction in inflammatory lesion count and no increase in abscesses or draining fistulas when compared with baseline) was achieved in 64% of patients receiving doxycycline 40 mg and 60% of patients receiving doxycycline 100 mg [10,42]. A phase II clinical trial compared minocycline 20 mg extended-release capsule once daily and 40 mg extended-release capsule once daily to doxycycline 40 mg once daily and placebo in the treatment of patients with mild-to-severe papulopustular rosacea for 16 weeks [40]. A higher proportion of patients who received the minocycline 40 mg extended-release capsule once daily achieved IGA treatment success (defined as an IGA grade 0 or 1 and ≥2-grade improvement) compared with placebo (66.04% vs. 11.54%; *p* < 0.0001), doxycycline 40 mg (66.04% vs. 33.33%; *p* = 0.0010), and minocycline 20 mg extended-release (66.04% vs. 31.91%; *p* = 0.007) [40]. Patients who received minocycline 40 mg extended-release capsule had a greater reduction in inflammatory lesions compared with placebo (−19.2 vs. −7.3; *p* < 0.0001), doxycycline 40 mg (−19.2 vs. −10.5; *p* = 0.0004), and minocycline 20 mg extended-release (−19.2 vs. −12.6; *p* = 0.0070) [40]. Minocycline 20 mg extended-release reduced lesion counts more than placebo (−12.6 vs. −7.3; *p* = 0.0290) [40]. While minocycline 40 mg has not been confirmed as a sub-antimicrobial dose, the minimal inhibitory concentration (MIC) for antimicrobial doses of minocycline is predicted to be the same as tetracycline and doxycycline [40]. Compared with the doxycycline MIC threshold of 1000 ng/mL as the limit for antimicrobial activity, the highest mean minocycline plasma levels achieved with the 40 mg extended-release formulation were both lower at 382.8 and 337.7 ng/mL on days 1 and 21 [40]. Overall, the minocycline 40 mg extended-release formulation was more efficacious than placebo and doxycycline 40 mg for rosacea at plasma concentrations predicted to be below the antimicrobial threshold [40].

## 3. Discussion

The serum concentration of oral tetracycline is decreased by about 50% when administered with food [24]. The absorption of oral doxycycline can decrease, remain unchanged, or increase when administered with food. The impact on the C_max_, the AUC, and the T_max_ varied with the formulation of doxycycline. When doxycycline hyclate delayed-release 100 mg and 150 mg tablets were administered with food, the C_max_ decreased by 24% and 19% and the AUC decreased by 13% and remained unchanged, respectively [22,23]. The bioavailability of the 200 mg tablet was not affected by administration with food, so the impact on absorption can vary with dose of the same formulation [22]. When doxycycline hyclate tablets and capsules were administered with food, the C_max_ decreased by 24% and 20% and the AUC decreased by 15% and remained unchanged, respectively [26]. The effect on bioavailability varied with formulation of tablet versus capsule, and capsules may be associated with better absorption of food. When doxycycline monohydrate capsules were administered with food, the T_max_ was delayed by an hour and twenty minutes, although the C_max_ increased by 7.5% and the AUC increased by 5.7% [27]. When doxycycline 40 mg capsules were administered with food, the C_max_ decreased by 45% and the AUC decreased by 22% [28]. 

The absorption of oral minocycline, depending on the dosage and formulation, can decrease, remain unchanged, or increase when administered with food. When minocycline hydrochloride tablets were administered with food, the C_max_ decreased by 12% and the T_max_ was delayed by 1.09 h [29]. When minocycline hydrochloride capsules were administered with food, the mean T_max_ was delayed by 1 h, but the extent of absorption was unchanged [14]. When minocycline hydrochloride extended-release tablets were administered with food, the C_max_, the T_max_, and the AUC increased [32]. When minocycline hydrochloride extended-release capsules were administered with food, the C_max_ and the AUC decreased [32]. The impact on the C_max_ and AUC varied with tablets versus capsules [32]. 

When sarecycline tablets were administered with food, the C_max_ decreased by 31% and the AUC by 27% [15]. Sarecycline is the only oral antibiotic where there was limited or no impact of food intake on clinical efficacy in pharmacokinetic studies [7]. No data exist for other tetracycline-class drugs in dermatology. Future research could examine the impact of taking doxycycline and minocycline with food on clinical efficacy. Low-dose doxycycline 20 mg formulations may be effective in treating adults with moderate facial acne [35,36]. Low-dose doxycycline 40 mg formulations may be effective in treating adults with moderate-to-severe rosacea and HS [10,37,38,39]. Minocycline 40 mg formulations may be effective in treating adults with mild-to-severe papulopustular rosacea [40]. 

## 4. Conclusions

If patients find that taking doxycycline, minocycline, and sarecycline with food optimizes tolerability and reduces gastrointestinal side effects (especially with doxycycline), this may merit consideration to encourage patient adherence. Since the impact of food intake on absorption varies with the formulation of doxycycline and minocycline, consulting the specific prescribing information (or package insert) may provide clinicians with the necessary information regarding dosage and administration, efficacy, and safety profile, to guide shared decision-making and optimize patient outcomes. Doxycycline and minocycline can cause gastrointestinal distress, esophagitis, esophageal ulceration, or idiopathic intracranial hypertension (pseudotumor cerebri) [14,22,23,26,27,29,30,31,32]. Administration with food can be helpful in reducing symptoms of gastrointestinal distress [14,22,23,26,27,29,30,31,32]. Based on FDA-approved United States package inserts, minocycline hydrochloride tablets and capsules and sarecycline tablets can be taken with or without food [14,15,29,30,31,32].

## 5. Materials and Methods

A literature search was performed using the PubMed database between February to May 2023. Key words included “tetracycline-class drugs”, “pharmacokinetics”, “absorption”, and “dermatology”. Inclusion criteria included articles written in English and relevant to tetracycline-class drugs absorption and efficacy. This search yielded a total of 131 articles published between 1977 to 2022, of which 29 met criteria for review. The package inserts for oral formulations of tetracycline, doxycycline, minocycline, and sarecycline were accessed on www.accessdata.fda.gov (accessed on 25 February 2023).

## Figures and Tables

**Table 1 antibiotics-12-01152-t001:** Summary of tetracycline-class drugs, recommendations, and absorption.

Brand and Generic Name/Formulation	Dosage	Half-Life	Package Insert Recommendations Regarding Food	Evidence for Decreased Absorption with Food Intake	Approved Dermatology Indication
Tetracycline hydrochloride [12].	250 and 500 mg capsules [12].	-	Food and some dairy products interfere with absorption [12].	Serum concentrations decreased by approximately 50% [24,25].	Adjunctive therapy for severe acne [12].
Doryx, Doxycycline hyclate delayed-release [22,23].	75, 80, 100, 150, 200 mg tablets [22,23].	-	If gastric irritation occurs, may be administered with food or milk [22,23].	The mean C_max_ was 24% lower (100 mg tablet) [22,23]. The mean AUC was 13% lower (100 mg tablet) [22,23]. The mean C_max_ was 19% lower (150 mg tablet) (unknown clinical significance) [22,23]. The AUC was unchanged (150 mg tablet) (unknown clinical significance) [22,23]. Bioavailability was unaffected by food (200 mg tablet) [22].	Adjunctive therapy for severe acne [22,23].
Acticlate, doxycycline hyclate [26].	150 mg tablets and capsules [26].	Range 18–22 h [26].	If gastric irritation occurs, may be given with food or milk [26].	The mean C_max_ was 24% lower (tablet) (unknown clinical significance) [26]. The mean AUC was 15% lower (tablet) (unknown clinical significance) [26]. The mean C_max_ was 20% lower (capsule) (unknown clinical significance [26]. The AUC was unchanged (capsule) [26].	Adjunctive therapy for severe acne [26].
Monodox, doxycycline monohydrate [27].	50, 75, and 100 mg capsules [27].	16.33 h [27].	If gastric irritation occurs, may be given with food [27].	Ingestion of a high-fat meal delayed the T_max_ by an average of 1 h and 20 min. However, in the same study, food enhanced the average C_max_ by 7.5% and the AUC by 5.7% [27].	Adjunctive therapy for severe acne [27].
Oracea, Doxycycline [28].	40 mg capsule [28].	21.2 h [28].	Take at least 1 h prior to or 2 h after meals [28].	The C_max_ decreased by 45% (40 mg dose) [28]. The AUC decreased by 22% (40 mg dose) [28].	Inflammatory lesions (papules and pustules) of rosacea in adult patients [28].
Dynacin, Minocycline hydrochloride [29].	50, 75, and 100 mg tablets [29].	11.38 to 24.31 h (average 17.03 h) [29].	May be taken with or without food [29].	Extent of absorption was decreased by 6% [29]. The C_max_ decreased by 12% and was delayed by 1.09 h when administered with food, compared to dosing under fasting conditions [29].	Adjunctive therapy of severe acne [29].
Minocin, Minocycline hydrochloride [14].	50 and 100 mg capsules [14].	11.1 to 22.1 h (average 15.5 h) [14].	May be administered with or without food [14].	Extent of absorption was unchanged compared with dosing under fasting conditions [14]. The mean T_max_ was delayed by 1 h when administered with food, compared to dosing under fasting conditions [14].	Adjunctive therapy for severe acne [14].
Minolira, Minocycline hydrochloride extended-release [30].	105 and 135 mg tablets [30].	15.6 h under fasting conditions and 17.1 h under fed conditions [30].	May be taken with or without food [30]. Ingestion with food may reduce risk of esophageal irritation and ulceration [30].	The C_max_ was 700 ng/mL under fasting conditions and 707 ng/mL under fed conditions [30]. The T_max_ was 2.0 h under fasting conditions and 3.5 h under fed conditions [30]. The AUC was 10,874 ng.hmL under fasting conditions and 12,000 ng.h/mL under fed conditions [30].	Inflammatory lesions of non-nodular moderate-to-severe acne vulgaris in patients 12 years of age and older [30].
Solodyn, Minocycline hydrochloride extended-release [31].	45, 55, 65, 80, 90, 105, 115, and 135 mg tablets [31].	-	May be taken with or without food [31]. Ingestion with food may reduce risk of esophageal irritation and ulceration [31].	Extent and timing of absorption when administered with a meal that contained dairy products did not differ from that of administration under fasting conditions [31].	Inflammatory lesions of non-nodular moderate-to-severe acne vulgaris in patients 12 years of age and older [31].
Ximino, Minocycline hydrochloride extended-release [32].	45, 90, and 135 mg capsules [32].	-	May be taken with or without food [32]. Ingestion with food may reduce risk of esophageal irritation and ulceration [32].	The AUC was 17.90 (5.56) mcg × h/mL under fasting conditions and 17.16 (3.19) mcg × h/mL when administered with a high-fat meal [32]. The C_max_ was 0.96 (0.32) mcg/mL under fasting conditions and 0.97 (0.25) mcg/mL when administered with a high-fat meal [32].	Inflammatory lesions of non-nodular moderate-to-severe acne vulgaris in patients 12 years of age and older [32].
SEYSARA, Sarecycline [15].	60, 100, and 150 mg tablets [15].	21–22 h [15].	May be administered with or without food [15].	The C_max_ decreased by 31% [15]. The AUC decreased by 27% [15].	Inflammatory lesions of non-nodular moderate-to-severe acne vulgaris in patients 9 years of age and older [15].

Abbreviations: C_max_ = maximal concentration; AUC = area under the curve; mg = milligrams; mL = milliliters; h = hour; mcg = micrograms; ng = nanograms.

**Table 2 antibiotics-12-01152-t002:** Low-dose doxycycline and minocycline efficacy data.

Treatment	Duration	Patient Population	Efficacy	Other Considerations
Doxycycline hyclate 20 mg tablets BID [35].	6 months [35].	Adults with moderate facial acne [35].	At 6 months: - reduced total comedones by 53.6% versus 10.6% in placebo group (*p* < 0.01) [35]. - reduced total inflammatory lesions by 50.1% compared to 30.2% in placebo group (*p* = 0.04) [35]. - reduced total lesions by 52.3% compared to 17.5% in placebo group (*p* < 0.01) [35].	-
Doxycycline hyclate 20 mg tablets BID [36].	3 months [36].	Adults with moderate facial acne [36].	At 3 months: - reduced number of papules by 84% [36]. - reduced number of pustules by 90% [36]. - statistical comparison was not performed between patients who received 20 mg versus 100 mg antimicrobial dosing [36].	Low-dose 20 mg doxycycline treatment was well tolerated [36]. One patient in the low-dose 20 mg doxycycline group dropped out due to nausea and vomiting [36]. Five patients in the antimicrobial dose 100 mg doxycycline group developed esophagitis [36].
Doxycycline 40 mg controlled-release capsules once daily [37].	16 weeks [37].	Adults with moderate-to-severe rosacea [37].	At 16 weeks: - reduced the total inflammatory lesion count by 11.8 in one study and 9.5 in the other study compared to 5.9 and 4.3, respectively, in the placebo arms (*p* < 0.001) [37].	-
Doxycycline monohydrate 40 mg capsules once daily [38].	16 weeks [38].	Adults with rosacea [38].	At 16 weeks: - had a greater reduction in inflammatory lesion count from baseline (−13.44 vs. −6.5; *p* = 0.0006), IGA score from baseline (−63.4% vs. −41%; *p* = 0.005), and erythema score from baseline (−1.3 vs. −0.7; *p* = 0.01) compared with topical metronidazole 1% gel monotherapy for 12 weeks plus placebo [38].	-
Doxycycline 40 mg delayed-release once daily [39].	16 weeks [39].	Adults with moderate-to-severe rosacea [39].	Both anti-inflammatory 40 mg delayed-release doxycycline and 100 mg doxycycline are equally effective once daily treatments for moderate-to-severe rosacea for up to 16 weeks [39].	5% of the subjects receiving 40 mg delayed-release doxycycline exhibited gastrointestinal symptoms compared to 26% of those receiving the 100 mg doxycycline dose [39].
Doxycycline 40 mg modified-release once daily [6,9,10].	12 weeks [6,9,10].	Adults with HS [6,9,10].	HiSCR was achieved in 64% of patients receiving doxycycline 40 mg and 60% of patients receiving doxycycline 100 mg [6,9,10].	-
Minocycline 20 mg extended-release capsule once daily and 40 mg extended-release capsule once daily [40].	16 weeks [40].	Adults with mild-to-severe papulopustular rosacea [40].	A higher proportion of patients who received the minocycline 40 mg ER formulation achieved IGA treatment success (defined as an IGA grade 0 or 1 and ≥2-grade improvement) compared with placebo (66.04% vs. 11.54%; *p* < 0.0001), doxycycline 40 mg (66.04% vs. 33.33%; *p* = 0.0010), and minocycline 20 mg ER (66.04% vs. 31.91%; *p* = 0.007) [40]. Patients who received 40 mg minocycline ER formulation had a greater reduction in inflammatory lesions compared with placebo (−19.2 vs. −7.3; *p* < 0.0001), doxycycline 40 mg (−19.2 vs. −10.5; *p* = 0.0004), and minocycline 20 mg ER (−19.2 vs. −12.6 *p* = 0.0070) [40].	-

Abbreviations: mg = milligrams; in = inches; IGA = investigators’ global assessment; HiSCR = Hidradenitis Suppurativa Clinical Response.

## Data Availability

No new data were created or analyzed in this study. Data sharing is not applicable to this article.

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
