# Peer review of "Oral Tetracycline-Class Drugs in Dermatology: Impact of Food Intake on Absorption and Efficacy"

_antibiotics, 2023, doi:10.3390/antibiotics12071152_

Round 1

Reviewer 1 Report

As attached

Reviewer 2 Report

The authors conducted a narrative review on the Impact of Food  Intake in Oral Tetracycline-Class Drugs

I have a few substantive questions/comments to consider:

The literature search period is too short and should be extended to at least one year.

The introduction is too long, part of this can be moved in the discussion section

The introduction should be concise, provide the necessary data to explain the reason for the research. Also a sentence about the reasons behind the review should be added

Reviewer 3 Report

I am very glad to review this interesting review. I have some comments for this manuscript.

1. The species of bacteria should be italicized.

2. The reference style should be consisted according to the guide for author. The title of references should not be in upper and lower case.

3.Sarecycline is the only oral antibiotic where population pharmacokinetic studies demonstrated limited or no impact of food intake on clinical efficacy. So, the context about sarecycline can be omitted. The manuscript described antibiotic and antiinflammatory dose of tetracyclin, doxycyclin and minomycin. Thus,I suggested the title should be replaced as "Various dose of Tetracyclin, Doxycyclin and Minomycin in Dermatology: Impact of Food  Intake on Absorption and Efficacy", which might be better.

Round 2

Reviewer 2 Report

The manuscript can be accepted